# Neuroprotective Effect of Bcl-2 on Lipopolysaccharide-Induced Neuroinflammation in Cortical Neural Stem Cells

**DOI:** 10.3390/ijms23126399

**Published:** 2022-06-07

**Authors:** Shin-Young Park, Joong-Soo Han

**Affiliations:** Biomedical Research Institute and Department of Biochemistry and Molecular Biology, College of Medicine, Hanyang University, 222 Wangsimni-ro, Seongdong-gu, Seoul 04763, Korea

**Keywords:** neural stem cells, lipopolysaccharide, B-cell lymphoma 2, neuronal differentiation, neuroinflammation

## Abstract

Neuroinflammation is involved in the pathogenesis of neurodegenerative diseases due to increased levels of pro-inflammatory cytokines in the central nervous system (CNS). Chronic neuroinflammation induced by neurotoxic molecules accelerates neuronal damage. B-cell lymphoma 2 (Bcl-2) is generally accepted to be an important anti-apoptotic factor. However, the role of Bcl-2 in neuroprotection against neuroinflammation remains to be determined. The purpose of this study was to investigate the neuroprotective effect of Bcl-2 on lipopolysaccharide (LPS)-induced neuroinflammation in cortical neural stem cells (NSCs). LPS decreased mRNA and protein levels of Tuj-1, a neuron marker, and also suppressed neurite outgrowth, indicating that LPS results in inhibition of neuronal differentiation of NSCs. Furthermore, LPS treatment inhibited Bcl-2 expression during neuronal differentiation; inhibition of neuronal differentiation by LPS was rescued by Bcl-2 overexpression. LPS-induced pro-inflammatory cytokines, including interleukin (IL)-6 and tumor necrosis factor alpha (TNF-α), were decreased by Bcl-2 overexpression. Conversely, Bcl-2 siRNA increased the LPS-induced levels of IL-6 and TNF-α, and decreased neuronal differentiation of NSCs, raising the possibility that Bcl-2 mediates neuronal differentiation by inhibiting the LPS-induced inflammatory response in NSC. These results suggest that Bcl-2 has a neuroprotective effect by inhibiting the LPS-induced inflammatory response in NSCs.

## 1. Introduction

Neuroinflammation is a contributing factor exacerbating the pathology of several neurodegenerative diseases, such as Alzheimer’s disease, Parkinson’s disease, and multiple sclerosis [1]. Neurodegeneration occurs in the central nervous system (CNS) through the loss of neuronal structure and function [2]. Recent studies have identified the lipopolysaccharide (LPS) as a causative agent of neurodegeneration [3]. It has been demonstrated that LPS-induced neuroinflammation triggers the primary damage caused by increased levels of pro-inflammatory cytokines, including tumor necrosis factor-α (TNF-α), interleukin-1β (IL-1β), and interleukin-6 (IL-6), in the brain of LPS-treated mice [4]. Pro-inflammatory cytokines lead to chronic neuroinflammation, which aggravates some pathological conditions and neuronal loss in the CNS [5]. Brains affected by neurodegenerative diseases have elevated levels of pro-inflammatory cytokines [6]. In addition, LPS-treated mice display increased neuronal death and impaired synaptic plasticity [7,8], suggesting that the regulation of LPS-induced neuroinflammation could be important in providing a potential strategy to reduce or prevent the development of neurodegenerative diseases.

Neural stem cells (NSCs) are self-renewing, multipotent cells that generate the neurons and glia of the nervous system during embryonic development [9]. NSCs can be isolated and cultured from primary cortical or hippocampal cultures in the presence of mitogenic growth factors, such as basic fibroblast growth factor (bFGF) and epidermal growth factor [10]. In neurodegenerative disease, NSCs contribute to the formation of new CNS cells [11]. Recently, NSCs transplantation experiments have challenged to injury area in the embryo and adult brain [11]. NSCs are directed to the damaged sites in CNS injury where they elicit proliferation and migration that results in the activation of a regenerative response [12]. Therefore, its putative clinical application has attracted considerable interest. During inflammatory events in the CNS, cytokines are released and affect NSC populations and their properties [12]. Although regeneration and protection of NSCs are important in neurodegenerative diseases, effects of neuroinflammation on neuronal differentiation of NSCs have not been studied yet.

B-cell lymphoma 2 (Bcl-2) is generally accepted as an important anti-apoptotic factor [13]. Bcl-2 inhibits cell apoptosis by enhancing the cell viability in response to different stimuli [14], acting in a central control point in the pathway to apoptotic cell death. While it is best known for its function in this regard, Bcl-2 has been implicated in neuronal differentiation and neurite outgrowth through multiple signaling pathways [15,16,17,18]. Furthermore, numerous studies suggest that Bcl-2 has an anti-inflammatory function in different cells [19,20]. Bcl-2 expression mediates LPS-induced neutrophilic inflammation and chronic airway inflammation [21], suggesting that Bcl-2 may have a role in LPS-induced neuroinflammation. However, the anti-inflammatory function of Bcl-2 in neuroinflammation of cortical NSCs has not been determined. In rats, neurogenesis peaks at embryonic day (E14) to E15 during embryonic cortical development [22]. Moreover, Bcl-2 is highly expressed in the brain, especially on E15, in the mouse neurogenesis stage [23,24], indicating that Bcl-2 may have a considerable impact on neuronal differentiation. Therefore, we examined the effect of Bcl-2 on LPS-induced neuroinflammation of NSCs on E15.

## 2. Materials and Methods

### 2.1. Materials

Coon’s modified Ham’s F-12 medium and human insulin were purchased from Sigma Chemical Co. (St Louis, MO, USA). Penicillin/streptomycin solution, neurobasal medium, and B27 were purchased from Gibco-Thermo Fisher Scientific (Waltham, MA, USA), and bFGF was purchased from R&D Systems (Minneapolis, MN, USA). LPS (*Escherichia. coli* O55:B5) was obtained from MilliporeSigma (St. Louis, MO, USA). The following antibodies were purchased: anti Bcl-2 (#SC-7382) from Santa Cruz Biotechnology (Dallas, TX, USA); anti β-tubulin type III (TUJ1) (#801202) from BioLegend (San Diego, CA, USA); and anti-calnexin (#ADI-SPA-860-F) from Enzo Life Sciences (Farmingdale, NY, USA). Fluorescein-(DTAF)-conjugated streptavidin (#016-010-084) was purchased from Jackson Immuno Research (West Grove, PA, USA). All other chemicals were of analytical grade.

### 2.2. Primary Culture of Neural Precursor Cells

All procedures using animals were performed according to the Hanyang University guidelines for the care and use of laboratory animals, and were approved by the Institutional Animal Care and Use Committee of Hanyang University (HY-IACUC-18-0028). Pregnant Sprague–Dawley (SD) rats were obtained from Orient Bio Inc. (Seoul, Korea). The brain cortex from E15 rat embryos (Sprague–Dawley rats) was mechanically triturated in Ca^2+^/Mg^2+^-free Hank’s balanced salt solution (Gibco), seeded at 2 × 10^5^ cells in 10-cm culture dishes (Corning Life Sciences, Corning, NY, USA), precoated with 15 μg/mL poly-L-ornithine (Sigma), and 1 μg/mL fibronectin (Invitrogen, Waltham, MA, USA). The cells were then cultured for 5–6 days in serum-free N2 medium supplemented with 20 ng/mL bFGF (R&D Systems Inc.). These neural precursor cells were induced to differentiate by withdrawing bFGF and incubating them in serum-free N2 medium. The cells generated by precursor cell proliferation were dissociated in Hank’s balanced salt solution and plated (3 × 10^4^ cells per well on coated 24-well plates, 3 × 10^5^ cells per well on coated 6-well plates, and 1 × 10^6^ cells per dish on coated 6-centimeter culture dishes). All experiments were performed using passage 1 (P1) neural stem cells.

### 2.3. Transient Transfection of NSCs

For the Bcl-2 knockdown experiments, Bcl-2 siRNA (5′-GAAUCAAGUGUUCGUCAUA-3′) was used (Bioneer, Daejeon, Korea), as well as control siRNA (On-target plus non-targeting pool; Dharmacon, Lafayette, CO, USA). Each siRNA (100 nM) was transiently transfected into NSCs using Lipofectamin RNAiMAX reagent (Invitrogen) according to the manufacturer’s protocol. For the Bcl-2 overexpression experiments, the NSCs were transiently transfected with 5 μg of either MSCV-IRES-EGFP or *rBcl-2*-MSCV-IRES-EGFP using a Nucleofector^TM^ Kit (Lonza, Basel, Switzerland) according to the manufacturer’s protocol.

### 2.4. Real-Time PCR and Reverse Transcription (RT)-PCR

The cDNA was prepared using the total mRNA extracted from the cells with TRIzol^®^ reagent (Thermo Fisher Scientific); 2-microgram samples of RNA were reverse-transcribed using random hexamer mixed primers. The cDNA thus formed was amplified by PCR using the following primers:

*Tuj1* (5′-ATGAGGGAGATCGTGCACATC-3′ and 5′-TGAATAGGTGTCCAAAGGCCC-3′),

*IL-6* (5′-TGATGGATGCTTCCAAACTG-3′ and 5′-GAGCATTGGAAGTTGGGGTA-3′),

*TNF-α* (5′-ACTGAACTTCGGGGTGATTG-3′ and 5′-GCTTGGTGGTTTGCTACGAC-3′)

*Gapdh* (5′-CTCGTCTCATAGACAAGATG-3′ and 5′-AGACTCCACGACATACTCAGCAC-3′).

For real-time PCR, 5 µL of RT reaction product was amplified in duplicate at a final volume of 20 µL iQ^TM^ SYBR^®^ Green Supermix (Bio-Rad Laboratories, Hercules, CA, USA). The PCR conditions were as follows: denaturation at 94 °C for 30 s, annealing at 58 °C for 1 min, and extension at 72 °C for 1 min. Thirty cycles were used for amplification of all cDNAs. The thermocycling conditions (*Tuj1*) were 95 °C for 10 min, followed by 40 cycles of 95 °C for 15 s and 60 °C for 1 min. The primer sequences for real-time PCR were the same as those used for RT-PCR. The PCR products were analyzed on a 1.5% agarose gel. All gene expression values were normalized to those of *Gapdh*.

### 2.5. Cytotoxicity of NSCs Exposed to LPS

NSCs (5 × 10^4^ cells/mL) were treated with LPS at 10 ng, 100 ng, 500 ng, 1 μg, 2 μg, or 5 μg/mL for 1 day in the absence of bFGF. The cells were collected for analysis for cell viability and performed using the Cell Counting Kit-8 (CCK-8; Sigma-Aldrich). Absorbance (450 nm) was measured using a microplate reader (BioRad Laboratories, Inc., Hercules, CA, USA); the experiment was performed in triplicate. Cytotoxic activity was calculated using the following formula: cytotoxicity (%) = (1 − A_450_ of target cells/A_450_ of control cells) × 100, where A_450_ = absorbance at 450 nm.

### 2.6. Western Blot Analysis

The cells were lysed in 20-millimolar Tris–HCl (pH 7.5) containing 150 mM NaCl, 1 mM ethylenediaminetetraacetic acid (EDTA), 1 mM ethylene glycol-bis (2-aminoethyl ether)-N,N,N’,N’-tetraacetic acid (EGTA), 2.5 mM sodium pyrophosphate, 1% Triton X-100, 1 mM phenylmethylsulfonyl fluoride (PMSF), and 1 mM Na_3_VO_4_. The protein samples (30 μg) were loaded on 12% SDS–polyacrylamide gels, electrophoresed, and transferred to nitrocellulose membranes (Amersham Pharmacia Biotech, Amersharm, UK) after electrophoresis. After blocking with 5% non-fat dried milk for 1 h, the membranes were incubated with primary antibodies (TUJ1 (1:2000 dilution), Bcl-2 (1:1000 dilution), calnexin (1:5000 dilution)), and then with the HRP-conjugated secondary antibodies Anti-mouse IgG (#7076) and Anti-rabbit IgG (#7074) at 1:2000 dilution (New England Biolabs, Ipswich, MA, USA). The specific bands were detected by ECL (Amersham Pharmacia Biotech).

### 2.7. Immunofluorescence Staining

The cells were washed with PBS and fixed with 4% paraformaldehyde in PBS, followed by three washes with PBS at room temperature. They were then permeabilized with 0.1% Triton X-100 in PBS for 10 min, followed by three washes with PBS, and then blocked with 10% normal goat serum in PBS containing 0.5% Tween 20 for 1 h at room temperature. Next, the cells were incubated with the mouse monoclonal anti-β-tubulin type III (TUJ1) primary antibody (1:1000 dilution) at 4 °C. The cells were then stained with streptavidin-conjugated secondary antibody (1:400 dilution) for 1 h before mounting with Vectashield (Vector Laboratories, Burlingame, CA, USA) containing 4,6-diamidino-2-phenylindole (DAPI). The immunoreactive cells were detected using a TCS SP5 confocal imaging system (Leica Microsystems, Wetzlar, Germany) at 50× magnification.

### 2.8. Measurement of Neurite Outgrowth

The cells were cultured on coverslips coated with fibronectin in 24-well plates, fixed with 0.1% (*w/v*) picric acid/PBS containing 4% (*w/v*) paraformaldehyde, and incubated overnight at 4 °C with β-tubulin type III (TUJ1) antibody (1:1000 dilution). After incubation with streptavidin-conjugated secondary antibody (1:400 dilution) for 1 h, the cells were mounted on slides with Vectashield. TUJ1-positive cells were photographed using the TCS SP5 confocal imaging system (Leica Microsystems), and the morphological characteristics were quantitated using ImageJ software (NIH; available online: http://rsb.info.nih.gov/ij/, accessed on 20 May 2021). The length of the primary neurite was defined as the distance from the soma to the tip of the longest branch. For each graph, the data on neurite length were generated from randomly selected areas in at least five independent cultures from three independent experiments, and more than 100 cells were counted for each condition in each experiment.

### 2.9. Enzyme-Linked Immunosorbent Assay (ELISA)

Cell supernatants were collected and tested for IL-6 using a rat IL-6 Quantikine^®^ ELISA kit (#R6000B, R&D Systems) or TNF-α using a rat TNF-α Quantikine^®^ ELISA kit (#RTA00, R&D Systems). Plates were read on a SpectraMax M2 microplate reader (Molecular Devices, Sunnyvale, CA, USA) and analyzed using SOFTmax analysis software (Molecular Devices).

### 2.10. Statistical Analysis

The statistical analysis was performed with IBM-SPSS software, version 18.0 (IBM, Armonk, NY, USA). The data were expressed as means ± SD from three independent experiments. The differences were analyzed using the one-way ANOVA with a post hoc Tukey’s analysis. *p* values < 0.05 were considered statistically significant.

## 3. Results

### 3.1. LPS-Induced Neuroinflammation Inhibits Neuronal Differentiation of NSCs

In order to determine the non-toxic concentration of LPS in NSCs, cells were treated with various concentrations of LPS in the absence of bFGF. As shown in Figure 1A, there was no cytotoxicity by showing a cell viability of 90.4% at 1 μg/mL of LPS. Therefore, we used this concentration in the following experiments. Neurite outgrowth is one of the typical morphological changes that occurs during neuronal differentiation. In order to investigate the effect of LPS on neurite outgrowth in NSCs, cells were treated with LPS (1 μg/mL) after bFGF removal. After 3 days, the cells were immunochemically stained using antibodies against a neuronal marker (Tuj1). Tuj1-stained cells fluoresce green. As shown in Figure 1B,C, LPS treatment significantly inhibited neurite outgrowth compared to bFGF removal cells (*p* < 0.05). The mRNA and protein levels of Tuj1 were also reduced by LPS treatment after bFGF removal (Figure 1D,E), indicating that LPS attenuates neuronal differentiation. It is widely accepted that LPS is related closely to neuroinflammation; therefore, we examined whether LPS leads to neuroinflammation in NSCs. As shown in Appendix A, the expression of pro-inflammatory cytokines *IL-6* and *TNF-α* increased from 100 ng/mL of LPS concentration, and their expressions markedly (*p* < 0.05) increased by LPS treatment (1 μg/mL) in the absence of bFGF. Taken together, these results indicate that LPS-induced neuroinflammation affects the neuronal differentiation of NSCs.

### 3.2. Bcl-2 Has a Neuroprotective Effect on LPS-Induced Neuroinflammation in NSCs

We previously reported that Bcl-2 is required for neuronal differentiation [15,17]. Bcl-2 has also been shown to influence neuronal apoptosis through neuroprotective functions [25]. Therefore, we investigated whether Bcl-2 has a protective effect on the LPS-induced neuroinflammation in NSCs. First, we showed that Bcl-2 expression increased during neuronal differentiation (Appendix A), whereas LPS inhibited Bcl-2 expression (Figure 2A,B). Moreover, we found that Tuj1 expression levels were increased by Bcl-2 overexpression compared to vector control cells in the absence of bFGF or with LPS (Figure 2C,D). Furthermore, Bcl-2 overexpression also enhanced neurite outgrowth compared to vector control cells in the absence of bFGF or with LPS. (Figure 2E,F). We also assessed the levels of pro-inflammatory cytokines in the Bcl-2-overexpressing NSCs, in order to confirm whether Bcl-2 could inhibit LPS-induced neuroinflammation. As shown in Figure 3A, Bcl-2 overexpression dramatically (*p* < 0.05) decreased *IL-6* and *TNF-α* mRNA levels compared to vector control cells in the absence of bFGF or with LPS. Furthermore, we showed that LPS-induced pro-inflammatory cytokines, IL-6 and TNF-α, were suppressed by Bcl-2 overexpression compared to vector control in the absence of bFGF with LPS (Figure 3B,C). These data suggest that Bcl-2 has a neuroprotective effect on LPS-induced neuroinflammation in NSCs by blocking the production of pro-inflammatory cytokines.

### 3.3. Knockdown of Bcl-2 Impairs Neuroprotective Function in LPS-Induced Neuroinflammation of NSCs

Next, we transfected control or Bcl-2 siRNA into NSCs to confirm the neuroprotective role of Bcl-2 in LPS-induced neuroinflammation. As shown in Figure 4A,B, Bcl-2 knockdown efficiently decreased the expression levels of Tuj1 compared to control siRNA in the absence of bFGF. Moreover, we found that the expression levels of Tuj1 reduced by LPS treatment were further inhibited by Bcl-2 depletion. Neurite length was also significantly (*p* < 0.05) reduced by Bcl-2 knockdown compared to control siRNA in the absence of bFGF with LPS (Figure 4C,D), indicating that Bcl-2 depletion impairs its neuroprotective function in LPS-induced neuroinflammation.

Moreover, we found that the LPS-induced expression levels of *IL-6* and *TNF-α* were increased by Bcl-2 depletion compared to control siRNA in the absence of bFGF with LPS (Figure 5A). Furthermore, Bcl-2 knockdown potentiated LPS-induced production of IL-6 and TNF-α compared to control siRNA in the absence of bFGF with LPS (Figure 5B,C). These findings investigate the Bcl-2 has an anti-inflammatory function in LPS-induced neuroinflammation of NSCs. Taken together we suggest that Bcl-2 plays a neuroprotective function by inhibiting neuronal damage in LPS-induced neuroinflammation of NSCs.

## 4. Discussion

In the present study, we first investigated the role of Bcl-2 in LPS-induced neuroinflammation in NSCs. Present results show that Bcl-2 overexpression promoted neuronal differentiation and decreased levels in IL-6 and TNF-α, whereas their depletion results in decreased neuronal differentiation and increased levels in IL-6 and TNF-α in LPS-stimulated NSCs. These findings suggest that Bcl-2 has a neuroprotective role through anti-inflammatory function in LPS-induced neuroinflammation in NSCs (Figure 5D). Neuroinflammation is implicated in the pathophysiology of several psychiatric and neurodegenerative disorders involving cognitive dysfunction and reduced neurogenesis, including Alzheimer’s disease, Parkinson’s disease, and depression [26]. Pro-inflammatory cytokines, such as IL-6, TNF-α, and IL-1β, negatively influence the differentiation of neural precursor cells [27,28,29], impairing neural function. In the CNS, IL-6, and TNF-α have been implicated in extensive inflammation and progressive neurodegeneration after ischemic and traumatic injuries [30,31,32]. These results indicate that IL-6 and TNF-α can cause chronic neuroinflammation and neuropathogenesis. In this study, we found that LPS stimulation increases the expression levels of IL-6 and TNF-α, reducing the neuronal differentiation of NSCs. However, increasing evidence shows that inflammatory cytokines can also stimulate neurite outgrowth and regeneration [33,34]. Recent studies suggested that IL-6 can stimulate the migration of cultured cortical neurons [35], and that TNF-α promotes neurogenesis and brain repair in response to brain injury and infection in cultured neural stem/progenitor cells [36]. Although most of these functions are observed in physiologically low concentrations of these inflammatory cytokines, not pathologically high concentrations in inflammation, these results suggest a novel physiological role of pro-inflammatory cytokines in the neuronal differentiation of NSCs. Thus, it is necessary to elucidate the pathophysiological mechanisms of these cytokines that result in the differentiation disorder of neural stem cells in the state of neuroinflammatory attack.

Recently, Bcl-2 has been found to mediate inflammation in the immune system organs [37,38] and in the brain [39,40]. Under pathophysiological conditions, Bcl-2 protects the cells from TNF-mediated apoptosis, and has an anti-inflammatory function in endothelial cells via NF-κB inhibition [41]. In this study, Bcl-2 overexpression reduced the LPS-induced expressions of *IL-6* and *TNF-α*, whereas Bcl-2 knockdown increased the expressions of *IL-6* and *TNF-α* in NSCs, indicating that Bcl-2 acts as an anti-inflammation regulator. Bcl-2 also influences neurogenesis in the CNS neuronal development [42,43]. We previously reported that Bcl-2 is required for neuronal fate determination in NSCs [17], and Bcl-2 overexpression induces neurite outgrowth via Bmp4/Tbx3/NeuroD1 cascade in H19-7 rat hippocampal precursor cells [15]. In this study, we found that overexpression of Bcl-2 promoted neurite outgrowth of NSCs even under LPS-treated conditions, whereas its knockdown inhibited neurite outgrowth. Although overexpression or knockdown of Bcl-2 alone did not significantly change pro-inflammatory cytokines, we suggest that the synergistic effect of differentiation signal (bFGF removal) and Bcl-2 overexpression or knockdown could contribute to the neuronal differentiation. These findings indicate that Bcl-2 expression in NSCs is very important for the neuronal differentiation of NSCs. Furthermore, we demonstrated that Bcl-2 has an anti-inflammatory function in LPS-treated NSCs, which plays a neuroprotective function by inhibiting loss of neural function in LPS-induced neuroinflammation of NSCs. On the other hand, the Bcl-2 family consists of anti-apoptotic members, such as Bcl-2 and Bcl-xL, and pro-apoptotic members, such as Bax and Bak. It is well known that Bax or Bak promotes neuronal apoptosis [44]. However, this study is not related to neuronal apoptosis, indicating that Bcl-2 is not a pro-apoptotic member.

In conclusion, the present study demonstrated a novel neuroprotective role of Bcl-2 in LPS-induced neuroinflammation of NSCs. Treatment with non-toxic LPS (1 μg/mL) induced pro-inflammatory cytokines and inhibited neuronal differentiation. Here, we found that the inhibition of neuronal differentiation by LPS was restored due to the anti-inflammatory action of Bcl-2. Taken together, Bcl-2 has a neuroprotective effect, since the loss of neuronal function caused by LPS-induced inflammatory cytokines was restored by Bcl-2. Therefore, we suggest that Bcl-2 could be a novel target to prevent neuropathological processes.

## Figures and Tables

**Figure 1 ijms-23-06399-f001:**
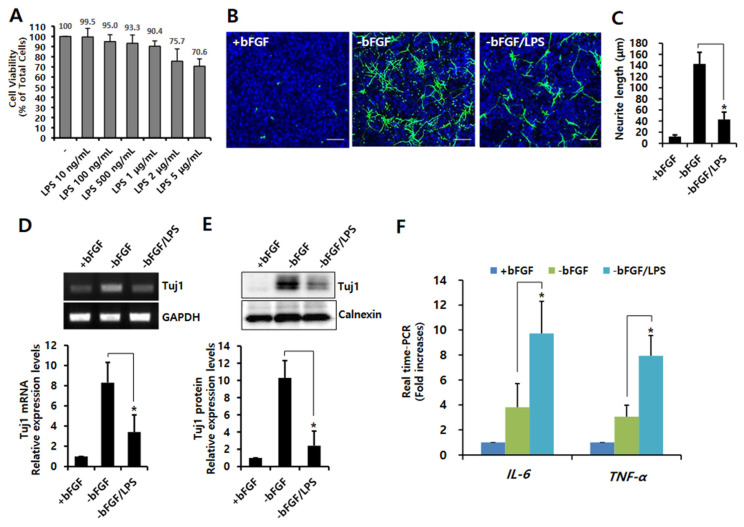
Effect of LPS stimulation on neuronal differentiation in NSCs. During the growth of isolated NSCs, bFGF was present (+bFGF, proliferation condition) to prevent differentiation and promote proliferation. In order to induce the neuronal differentiation, bFGF was removed (-bFGF, differentiation condition). (**A**) Cell viability in LPS-treated NSCs. NSCs were treated with LPS at 10 ng, 100 ng, 500 ng, 1 μg, 2 μg, or 5 μg/mL for 1 day in the absence of bFGF. The cells were collected for analysis for cell viability and performed using the Cell counting Kit-8. Absorbance was measured at 450 nm, and all experiments were performed in triplicate. Cytotoxic activity is expressed as the percentage of cell viability using the formula: cytotoxicity (%) = (1 − A450 of target cells/A450 of control cells) × 100; *n* = 3. Data are shown as means ± SD. (**B**,**C**) NSCs were treated with LPS (1 μg/mL) for 3 days in the absence of bFGF. They were stained with anti-Tuj1 (green) and DAPI (blue). Scale bar: 100 μm. (**C**) Neurite lengths were measured in randomly selected fields using three independent experiments, with *n* = 3 per group. Data are shown as means ± SD. * *p* < 0.05 compared with –bFGF control. (**D**) The cells were treated with LPS (1 μg/mL) for 12 h in the absence of bFGF. The Tuj1 mRNA level was analyzed using RT-PCR (upper panel) and real time-RT-PCR (graph); *n* = 3. Data are shown as means ± SD. * *p* < 0.05 compared with–bFGF control. (**E**) The cells were treated with LPS (1 μg/mL) for 1 day in the absence of bFGF. Western blotting was performed using anti-Tuj1 or anti-calnexin antibodies to detect the respective protein bands (upper panel). Band intensity (graph) was quantified with Quantity Ones^®^ software. Data are shown as means ± SD. * *p* < 0.05 compared with –bFGF control. (**F**) The cells were treated with LPS (1 μg/mL) for 12 h in the absence of bFGF. The mRNA levels of IL-6 and TNF-α were analyzed by real-time RT-PCR; *n* = 3. Data are shown as means ± SD. * *p* < 0.05 compared with –bFGF control, for IL-6 and TNF-α, respectively. Statistical significances were assessed by one-way ANOVA with a post hoc Tukey’s test.

**Figure 2 ijms-23-06399-f002:**
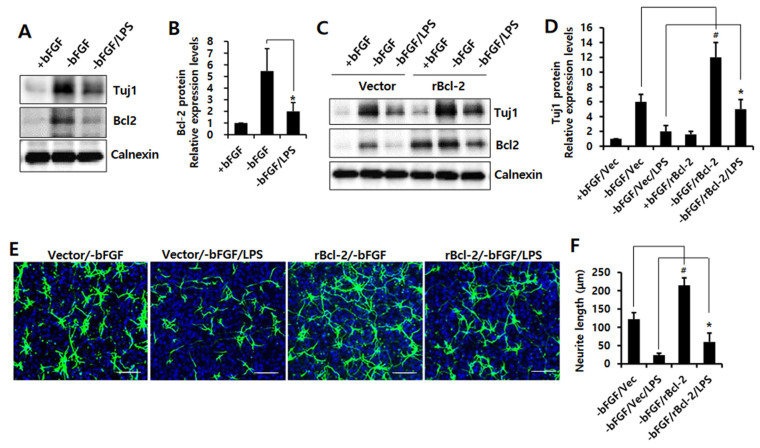
Effect of Bcl-2 overexpression on LPS-induced neuronal damage in NSCs. (**A**,**B**) NSCs were treated with LPS (1 μg/mL) for 1 day in the absence of bFGF. (**A**) Western blotting was performed using anti-Tuj1, anti-Bcl-2, or anti-calnexin antibodies to detect the respective protein bands. (**B**) Band intensity was quantified with Quantity Ones^®^ software. Data are shown as means ± SD. * *p* < 0.05 compared with –bFGF control. (**C**,**D**) MSCV-IRES-EGFP or *rBcl-2*-MSCV-IRES-EGFP was transfected into the NSCs for 48 h, then the cells were treated with LPS (1 μg/mL) for 1 day in the absence of bFGF. (**C**) Western blotting was performed using anti-Tuj1, anti-Bcl-2, or anti-calnexin antibodies to detect the respective protein bands. (**D**) Band intensity was quantified with Quantity Ones^®^ software. Data are shown as means ± SD. * *p* < 0.05 compared with –bFGF/Vector/LPS control. ^#^
*p* < 0.05 compared with –bFGF/Vector control. (**E**,**F**) MSCV-IRES-EGFP or *rBcl-2*-MSCV-IRES-EGFP was transfected into the NSCs for 48 h, and the cells were treated with LPS (1 μg/mL) for 3 days in the absence of bFGF. The NSCs were stained with anti-Tuj1(green) and DAPI (blue). Scale bar: 100 μm. (**F**) Neurite lengths were measured in randomly selected fields using three independent experiments; *n* = 3 per group. Data are shown as means ± SD. * *p* < 0.05 compared with –bFGF/Vector/LPS control. ^#^
*p* < 0.05 compared with –bFGF/Vector control. Statistical significances were assessed by one-way ANOVA with a post hoc Tukey’s test.

**Figure 3 ijms-23-06399-f003:**
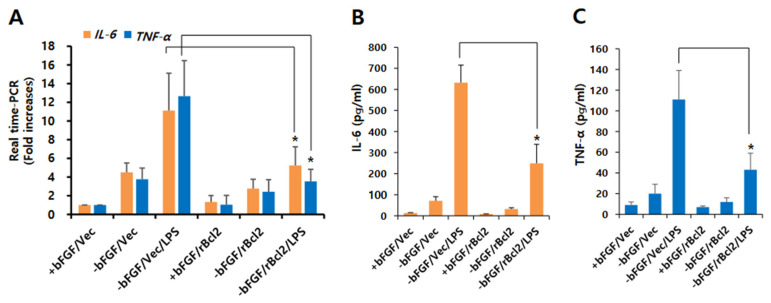
Effect of Bcl-2 overexpression on LPS-induced IL-6 and TNF-α production in NSCs. (**A**) MSCV-IRES-EGFP or *rBcl-2*-MSCV-IRES-EGFP was transfected into the NSCs for 48 h, and the cells were treated with LPS (1 μg/mL) for 12 h in the absence of bFGF. The mRNA levels of *IL-6* and *TNF-α* were analyzed by real-time RT-PCR; *n =* 3. Data are shown as mean ± SD. * *p* < 0.05 compared with –bFGF/Vector/LPS control, for *IL-6* and *TNF-α* respectively. (**B**,**C**) MSCV-IRES-EGFP or *rBcl-2*-MSCV-IRES-EGFP was transfected into the NSCs for 48 h, and the cells were treated with LPS (1 μg/mL) for 24 h in the absence of bFGF. The levels of IL-6 (**B**) and TNF-α (**C**) were measured by ELISA. Data are shown as means ± SD. * *p* < 0.05 compared with –bFGF/Vector/LPS control. Statistical significances were assessed by one-way ANOVA with a post hoc Tukey’s test.

**Figure 4 ijms-23-06399-f004:**
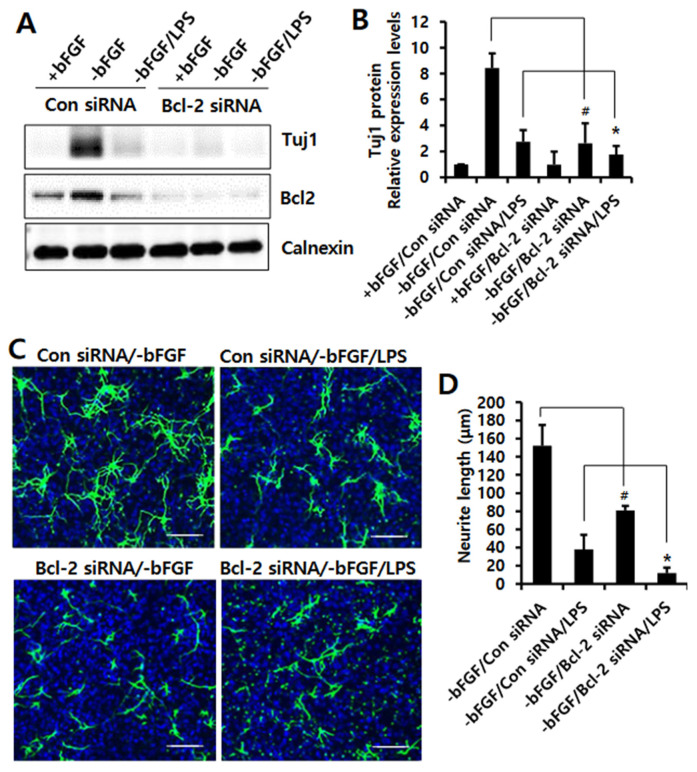
Effect of Bcl-2 depletion on LPS-induced neuronal damage in NSCs. (**A**,**B**) Control siRNA or Bcl-2 siRNA was transfected into the NSCs for 72 h, and then the cells were treated with LPS (1 μg/mL) for 1 day in the absence of bFGF. (**A**) Western blotting was performed using anti-Tuj1, anti-Bcl-2, or anti-calnexin antibodies to detect the respective protein bands. (**B**) Band intensity was quantified with Quantity Ones^®^ software. Data are shown as means ± SD. * *p* < 0.05 compared with –bFGF/control siRNA/LPS. ^#^
*p* < 0.05 compared with –bFGF/control siRNA. (**C**,**D**) Control siRNA or Bcl-2 siRNA was transfected into the NSCs for 72 h, and then the cells were treated with LPS (1 μg/mL) for 3 days in the absence of bFGF. The NSCs were stained with anti-Tuj1(green) and DAPI (blue). Scale bar: 100 μm. (**D**) Neurite lengths were measured in randomly selected fields using three independent experiments; *n* = 3 per group. Data are shown as means ± SD. * *p* < 0.05 compared with –bFGF/control siRNA/LPS. ^#^
*p* < 0.05 compared with –bFGF/control siRNA. Statistical significances were assessed by one-way ANOVA with a post hoc Tukey’s test.

**Figure 5 ijms-23-06399-f005:**
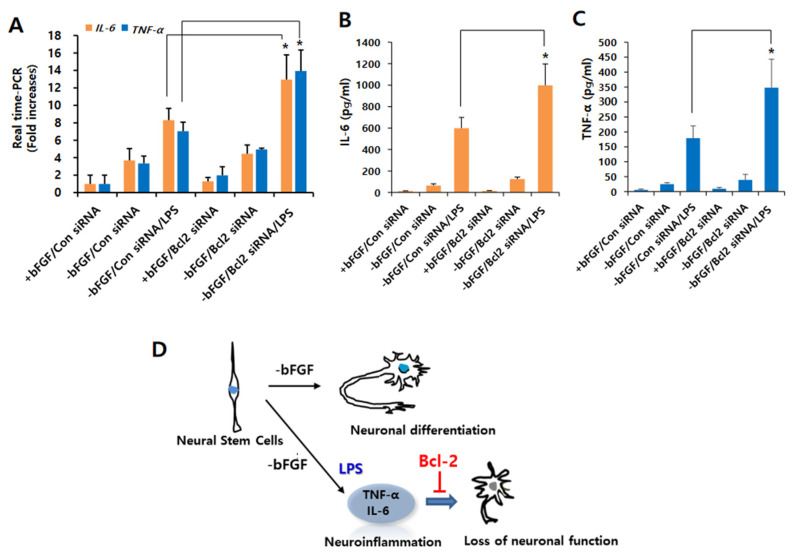
Effect of Bcl-2 depletion on LPS-induced IL-6 and TNF-α production in NSCs. (**A**) Control siRNA or Bcl-2 siRNA was transfected into the NSCs for 72 h, and then the cells were treated with LPS (1 μg/mL) for 12 h in the absence of bFGF. The mRNA levels of *IL-6* and *TNF-α* were analyzed by real-time RT-PCR; *n =* 3. Data are shown as means ± SD. * *p* < 0.05 compared with –bFGF/control siRNA/LPS, for *IL-6* and *TNF-α,* respectively. (**B**,**C**) Control siRNA or Bcl-2 siRNA was transfected into the NSCs for 72 h, and then the cells were treated with LPS (1 μg/mL) for 24 h in the absence of bFGF. The levels of IL-6 (**B**) and TNF-α (**C**) were measured by ELISA. Data are shown as means ± SD. * *p* < 0.05 compared with –bFGF/control siRNA/LPS. (**D**) The proposed model for Bcl-2-mediated neuronal differentiation in LPS-treated NSCs. The model suggests that Bcl-2 plays a neuroprotective role in LPS-induced neuroinflammation of NSCs, resulting in neuronal differentiation. Statistical significances were assessed by one-way ANOVA with a post hoc Tukey’s test.

## Data Availability

The data that support the findings of this study are available on request from the corresponding author.

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
