# Peer review of "Neuroprotective Effect of Bcl-2 on Lipopolysaccharide-Induced Neuroinflammation in Cortical Neural Stem Cells"

_ijms, 2022, doi:10.3390/ijms23126399_

Round 1
Reviewer 1 Report
The manuscript by Park and Han entitled „Neuroprotective effect of Bcl-2 on lipopolysaccharide-induced neuroinflammation in cortical neural stem cells” describes the role of Bcl-2 in neuronal stem cells. Treatment with LPS decreased mRNA and protein levels of neuron marker, and also suppressed neurite outgrowth. Targeting of Bcl-2 inhibited the LPS induced effects.
This topic is very relevant, however the study design is flawed in parts and interpretations are overstated. The explanation of same experimental design is missing, making it difficult to interpret the results.
Some points of criticism include:
1) Authors wrote : “B-cell lymphoma 2 (Bcl-2) is generally accepted as an important anti-apoptotic factor.” However it is well known that Bcl-2 regulate cell death, by either inhibiting or inducing apoptosis. (Kahle et al 2017; Garcia Saez et al 2020). Discussion of this ambivalence is missing.
2) LPS concentration is unphysiologically high. How was it chosen? 10 ng/mL of LPS is optimal for inducing pro-inflammatory responses without toxicity (Sivagnanam et al., 2010; Lively and Schlichter, 2013).
3) Since the authors are talking about cell death, it is not clear why cell death and cell number was not assessed in this study.
4) Since the authors write Bcl-2 expression is increased during neuronal differentation it is necessary to examine different time points
5) How were the incubation times chosen? Why are they different for different experimental settings?
Fig 1: LPS treatment for 3 days or 12h or for 1 day
Fig 2: LPS treatment for 3 days or 1 day etc.
Authors have to do a more systematic and consistent analysis of the temporal evolution.
6) Characterization of NSC is missing and explanation why E15 cultures were used, as well?
7) It is not clear why the used markers, like Tuj 1, etc., but not further established neuronal and glial markers were chosen. Authors have to use additional, well characterized markers.
8) The representative images do not match the quantification.
Author Response
I’d like to thank you for your helpful suggestions.
In response to your request, the reviewer’s suggestions were written in blue color, and we replied to every suggestion point by point, and indicated in the text by page and line numbers.
Please see the attachment.
Thank you so much.

Reviewer 2 Report
The authors mention that LPS-induced neuroinflammation affects neuronal damage and neuronal development via Bcl-2, and their results are very interesting. However, the conclusions are littered with claims based on cited literature, and the data in this study alone are not considered novel. If some of the critical improvements listed below can be addressed, I believe that the manuscript is ready for publication in this journal.
Comment 1
I do not know the population of primary culture cells used in this study. Depending on the population, the concept of this study may change. Data should be added that shows the population at each point in this analysis.
Comment 2
In Figures 1, 2 and 4, the authors should add data analyzed for cell death. If possible, the authors should also analyze the neuronal differentiation lineage in addition to cell death; Tuj-1 is a marker protein for juvenile neurons. Therefore, the decrease in Tuj-1-positive cells could be due to various possibilities, such as suppression of neuronal differentiation, promotion of differentiation into non-neuronal cells or reduction of cell survival. If possible, it would be better to analyze the differentiation lineage in addition to cell death. Regarding Figures 2 and 4, since Bcl-2 has an anti-apoptotic effect, it is not clear whether the Bcl-2 results are the result of its effect on cell survival or on neuronal maturation.
Comment 3
Regarding Figure 3, overexpression of Bcl-2 in the absence of LPS does not alter IL-6 or TNF-α. In contrast, Bcl-2 overexpression in the absence of LPS results in both Tuj-1 positivity and neurite outgrowth. A similar phenomenon occurs in Figure 5. The authors should explain this difference in as much detail as possible.
Comment 4
If the authors are pursuing a link to neuroinflammation, they should consider whether Bcl-2 has a similar effect as in the present study, not only on LPS-induced inflammation but also on inflammation caused by other factors. I think it needs to be clarified whether the inhibition of LPS-induced neurotoxicity by Bcl-2 is targeted to inflammatory cytokines or not.
Comment 5
The authors ask that you confirm that the statistical analysis of the graphs presented in this paper is correct. Student's t-test should only be used between two groups. Also, it seems to me that some combinations in one graph have statistical analysis done and some do not. For example, Con siRNA and Bcl-2 siRNA (and Bcl-2 siRNA and Bcl-2 siRNA/LPS) in Figure 4D are clearly significantly different.
Author Response

(The authors gave the same response as above.)

Reviewer 3 Report
Park and Han have conducted an interesting study and have found that Bcl-2 protects NSC-derived neurons from LPS-induced neuroinflammation in cell culture. In general terms, this is a well performed study that requires minor clarifications in the Methods sections.
1. Please, add to the manuscript the name of the companies from where primary and secondary antibodies were purchased, and write down the dilutions used for immunofluorescence and Western blot.
2. The statistical analysis applied (unpaired Student´s t test or one-way ANOVA) should be included in each figure legend.
Author Response

(The authors gave the same response as above.)

Round 2
Reviewer 1 Report
The authors adressed all my points of critisism.
Author Response
Dear, Reviewer 1
Thanks to your wonderful comments, our paper has been greatly improved.
Thank you again.
Sincerely
Shin-Young Park & Joong-Soo Han
Reviewer 2 Report
Although this manuscript is insufficient in terms of novelty, the research and logic of the manuscript have reached a sufficient level for publication. However, the authors should revise one point as follows.
Not all figures are included in the edited manuscript. All additional figures (cell viability and immunostaining) listed in the cover letter should be added within the manuscript.
Author Response
Dear, Reviewer 2
Thanks to your wonderful comments, our paper has been greatly improved.
Please see the attachment.
Thank you again.
Sincerely
Shin-Young Park & Joong-Soo Han
